# The genetic basis of dermatophytosis skin infection susceptibility

Hele Haapaniemi [1], Reyhane Eghtedarian[1], Anniina Tervi [1], Jesse Valliere[2,3], Estonian Biobank Research Team*, FinnGen*, Erik Abner [4] & Hanna M. Ollila [1,2,3,5] ✉

Dermatophytosis is a fungal infection affecting keratinized tissues such as skin, nails, and hair, presenting as red and itchy patches, nail thickening, or hair loss. It affects around 20% of the global population but the genetic architecture remains poorly understood. We performed a genome-wide association meta-analysis of over 250,000 cases and 1.37 million controls from FinnGen, Estonian Biobank, UK Biobank, and the Million Veteran Program and identified 30 genome-wide significant loci, including seven missense variants and two loci in high linkage disequilibrium with missense variants. Top associations were near *ZNF646*, *HLA-DQB1*, *FLG*, *FTO*, *SLURP2*, and *KRT77*. Additionally, dermatophytosis subtype analyses revealed 44 signals. Our results highlight the role of disrupted keratin biology, skin barrier defects, immune dysfunction, and obesity in dermatophytosis risk. We also observed genetic overlap with other skin conditions and obesity-related traits, providing insights into disease mechanisms and potential targets for prevention and treatment.

Dermatophytosis, commonly known as ringworm, is a prevalent fungal infection affecting the skin, hair, and nails. It is caused by dermatophytes, a group of keratinophilic fungi with the unique ability to utilize keratin, a structural protein in the outer layer of human skin, as a nutrient source, leading to various clinical symptoms[1,2]. While typically limited to the outermost layers of the skin, dermatophytosis can become more severe in certain patient groups. For example, in immunocompromised and diabetic patients with compromised immune system or skin barrier, the infection may invade deeper layers of the skin and can lead to severe and invasive disease[3].

The global incidence of dermatophytosis has made it a significant public health concern, particularly in regions with warm and humid climates that favor fungal growth. It has been estimated that around 20–25% of people are infected with dermatophytes at some point in their lives, and the incidence rate is constantly rising[4]. The prevalence varies between continents and countries, but more recent studies from European countries have estimated prevalence rates around 12–17%[5,6]. Dermatophytosis skin infections can develop to people of all ages.

Infections of the scalp (tinea capitis), are more common in children, while other tineas more commonly affect post pubertal individuals with a peak at mid-age[7,8]. Besides geographic location and age, the individual vulnerability to dermatophytosis depends on several factors, including sex, season, socioeconomic status, personal hygiene and cultural conditions[7,8]. In addition, existing skin diseases or skin lesions, together with immunocompromising factors, can affect the susceptibility to dermatophytosis.

Dermatophytosis presents a wide range of symptoms, depending on the site of infection. On the skin (e.g., tinea corporis, tinea pedis), it typically manifests as red, scaly, and itchy patches that often form a ring-like pattern, hence the name "ringworm". Infections of the scalp (tinea capitis) can lead to hair loss and inflammation, while nail infections (onychomycosis) cause thickening, discoloration, and brittleness of the nails[9]. The infection is highly contagious and can spread through direct contact with infected individuals or animals, as well as indirectly through contaminated objects like clothing, towels, and grooming tools[10].

[1]Institute for Molecular Medicine Finland, FIMM, HiLIFE, University of Helsinki, Helsinki, Finland. [2]Program in Medical and Population Genetics, Broad Institute of Harvard and MIT, Cambridge, MA, USA. [3]Center for Genomic Medicine, Massachusetts General Hospital, Boston, MA, USA. [4]Functional and Population Genomics, Institute of Genomics, University of Tartu, Tartu, Estonia. [5]Department of Anesthesiology, Mass General Brigham, Harvard Medical School, Boston, MA, USA. *Lists of authors and their affiliations appear at the end of the paper. ✉e-mail: hanna.m.ollila@helsinki.fi

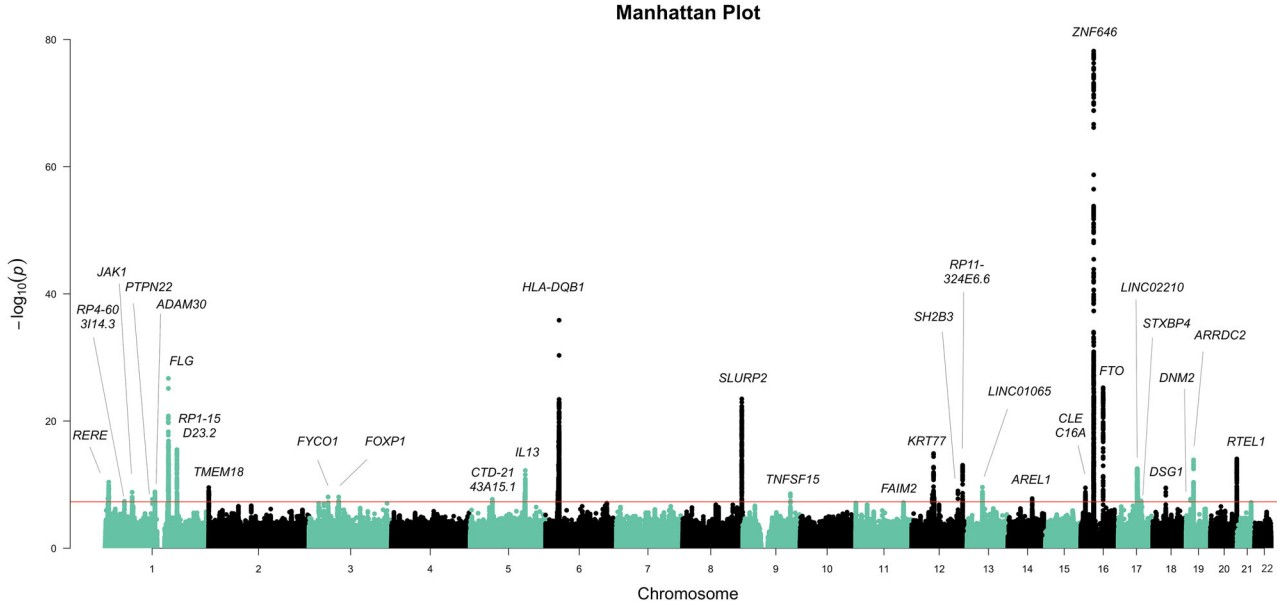

**Fig. 1 | Manhattan plot for dermatophytosis infection meta-analysis.** The meta-analysis included 256,822 cases and 1,372,501 controls from FinnGen, EstBB, UKB, and MVP. The X-axis represents the chromosomal position of each genetic variant, and the Y-axis shows the $-\log_{10}(p$-value) from the association test. Association analyses were conducted using logistic regression under an additive genetic model.

A two-sided Wald test was used to evaluate the null hypothesis that the effect size (beta) equals zero. The horizontal line indicates the genome-wide significance threshold ($p < 5 \times 10^{-8}$), which accounts for multiple hypothesis testing across the genome. The summary statistics used to generate this figure are available at the GWAS catalog.

While environmental factors associate with dermatophytosis infections, genetic studies provide an avenue to understand the biological mechanisms that contribute to risk and the development of dermatophytosis. Here, we aimed to understand host factors that affect susceptibility to dermatophytosis infections by performing the largest genome-wide association analysis with over 250,000 dermatophytosis cases from FinnGen, the UK Biobank (UKB), the Estonian biobank (EstBB) and the Million Veteran Program (MVP). Our findings highlight the role of barrier organs and variety of immune functions in the development of dermatophytosis.

## Results

### GWAS shows an association between dermatophytosis and 30 genetic loci

To explore the host genetic components contributing to dermatophytosis, we performed GWAS and meta-analysis in FinnGen ($N = 27,662$ cases and 471,729 controls), UKB ($N = 27,755$ cases and 380,368 controls), EstBB ($N = 50,241$ cases and 106,586 controls) and Million Veterans Program ($N = 151,164$ cases and 413,818 controls).

With data from 256,822 dermatophytosis cases and 1,372,501 controls, we identified 30 genome-wide significant loci ($p < 5 \times 10^{-8}$) associated with dermatophytosis infection (Fig. 1; Table 1, and Supplementary Data Table S1). The most significant loci were *ZNF646* ($p = 6.60 \times 10^{-79}$), *HLA* ($p = 1.42 \times 10^{-36}$), *FLG* ($p = 1.96 \times 10^{-27}$), *SLURP2* ($p = 3.33 \times 10^{-24}$) and *KRT77* ($p = 1.28 \times 10^{-15}$).

Dermatophytosis can be classified into subtypes based on the region of the body where the infection is manifesting. To understand the subtype-specific associations, we additionally ran GWAS on each of these subtypes in FinnGen, EstBB, and UKB, along with the publicly available genome-wide association data for subtypes in the MVP (Fig. 2). The association statistics for each of our lead variants in these subtypes are reported in Supplementary Data S9 (case and control numbers are presented in S8). We show association of several lead variants from the main dermatophytosis analysis at the subtype level ($p < 0.0002$, Bonferroni corrected threshold), especially in subtypes B35.1 (nail ringworm), B35.3 (athlete's foot) and B35.4 (dermatophytosis of the body). When studying the lead variants at the subtype level,

we see a consistent pleiotropic association with a subset of variants across several different subtypes, including *FLG, KTR77, ZNF646*, and *FTO* at genome-wide significant level ($p < 5 \times 10^{-8}$), highlighting the role of these genes in the defense against dermatophytosis infections. In addition, we report associations in another zinc finger gene *ZNF668* (B35.1 and B35.3) located in proximity of the *ZNF646 gene*. Furthermore, we report 34 previously unreported associations, such as rs112088479 closest to the *NOTCH2* gene that is linked to skin homeostasis and epidermal differentiation[11,12]. We present Manhattan plots for each subtype in Supplementary Figs. and the lead variants in Supplementary Data S10.

The majority of the genetic associations of complex diseases are regulatory variants typically located at the noncoding or intronic regions of the genome, and usually, these associations affect gene expression levels rather than have direct impact on protein structure[13]. In our meta-analysis, nine out of thirty variants were either missense variants or in high LD (linkage disequilibrium) with a missense variant (http://www.mulinlab.org/vportal/index.html). (Table 2).

The missense variants with the most significant association with dermatophytosis infection were identified within the *ZNF646* gene (rs7196726, beta = 0.066, $p = 6.60 \times 10^{-79}$) and its neighboring genes *PRSS3* (rs35713203, beta = 0.065, $p = 1.90 \times 10^{-78}$) and *HSD3B7* (rs9938550, beta = 0.056, $p = 3.43 \times 10^{-57}$). In addition, we identified highly significant missense variants ($<5 \times 10^{-15}$) within the *FLG* gene (rs558269137, beta = −0.222, $p = 1.96 \times 10^{-27}$) and the *KRT1* gene (rs14024, beta = −0.030, $p = 3.95 \times 10^{-15}$). All of the reported missense variants have minor allele frequency above 1% and all of them are predicted to be benign based on their Polyphen score ($<0.15$), estimating the impact of an amino acid substitution on the structure and function of a human protein[14].

While missense variation can indicate a likely causal gene at the locus, the majority of the associated variants were located in noncoding or intronic regions and likely contribute to disease risk by affecting gene expression. To elucidate possible affected genes near the regions of the strongest non-missense associations (rs10094888 closest to *SLURP2* and rs1794269 closest to *FTO*), we performed a colocalization analysis with expression data from GTEx[15] (https://

**Table 1 | Genome-wide significant lead variants from dermatophytosis infection meta-analysis using data from FinnGen, EstBB, UKB and MVP comprising 256,822 cases and 1,372,501 controls**

| SNP | CHR | Effect allele | Non-effect allele | AF | P | OR (95% CI) | Direction | Nearest gene | Most severe consequence |
|---|---|---|---|---|---|---|---|---|---|
| rs6577497 | 1 | A | T | 0.640 | $4.05 \times 10^{-11}$ | 1.024 [1.016, 1.032] | ++++ | RERE | Intron variant |
| rs56733721 | 1 | A | G | 0.605 | $4.24 \times 10^{-08}$ | 0.980 [0.972, 0.987] | ---- | RP4-603I14.3 | Downstream gene variant |
| rs506025 | 1 | A | G | 0.839 | $1.58 \times 10^{-09}$ | 0.970 [0.961, 0.980] | ---- | JAK1 | Intergenic variant |
| rs2476601 | 1 | A | G | 0.111 | $2.03 \times 10^{-08}$ | 1.032 [1.02, 1.044] | ++++ | PTPN22 | Missense variant |
| rs35273427 | 1 | T | C | 0.910 | $1.40 \times 10^{-09}$ | 0.962 [0.951, 0.974] | ---- | ADAM30 | Missense variant |
| rs558269137 | 1 | CACTG | C | 0.984 | $1.96 \times 10^{-27}$ | 0.800 [0.768, 0.834] | ?--- | FLG FLG-AS1 | Frameshift variant |
| rs10912488 | 1 | A | G | 0.730 | $3.16 \times 10^{-16}$ | 1.032 [1.024, 1.040] | ++++ | RP1-15D23.2 | Intron variant |
| rs60383093 | 2 | G | GTTC | 0.829 | $2.90 \times 10^{-10}$ | 1.032 [1.022, 1.042] | +?++ | TMEM18 | Intergenic variant |
| rs11130078 | 3 | T | C | 0.409 | $8.73 \times 10^{-09}$ | 0.979 [0.971, 0.986] | ---- | FYCO1 | Intron variant |
| rs62246017 | 3 | A | G | 0.281 | $8.86 \times 10^{-09}$ | 0.977 [0.969, 0.985] | ---- | FOXP1 | Intron variant |
| rs37807 | 5 | A | T | 0.684 | $2.05 \times 10^{-08}$ | 0.979 [0.971, 0.986] | ---- | CTD-2143A15.1 | Intergenic variant |
| rs848 | 5 | A | C | 0.295 | $5.95 \times 10^{-13}$ | 0.972 [0.964, 0.980] | ---- | IL13 TH2LCRR | 3 prime UTR variant |
| rs1794269 | 6 | T | C | 0.401 | $1.42 \times 10^{-36}$ | 1.047 [1.038, 1.055] | ++++ | HLA-DQB1 | Intergenic variant |
| rs10094888 | 8 | T | C | 0.362 | $3.33 \times 10^{-24}$ | 1.036 [1.028, 1.044] | ++++ | SLURP2 | Upstream gene variant |
| rs56069985 | 9 | A | G | 0.944 | $2.80 \times 10^{-09}$ | 1.05 [1.033, 1.066] | ++++ | TNFSF15 | Upstream gene variant |
| rs640081 | 12 | A | G | 0.659 | $3.89 \times 10^{-09}$ | 1.022 [0.828, 1.260] | ++++ | FAIM2 | Intron variant |
| rs1829637 | 12 | A | G | 0.303 | $1.28 \times 10^{-15}$ | 1.031 [1.023, 1.039] | ++++ | KRT77 | Intron variant |
| rs3184504 | 12 | T | C | 0.419 | $9.74 \times 10^{-10}$ | 1.022 [1.014, 1.030] | ++++ | SH2B3 | Missense variant |
| rs612057 | 12 | T | C | 0.455 | $8.79 \times 10^{-14}$ | 1.026 [1.018, 1.034] | ++++ | RP11-324E6.6 | Intron variant |
| rs1885767 | 13 | A | G | 0.398 | $2.56 \times 10^{-10}$ | 1.023 [1.015, 1.031] | ++++ | LINC01065 | Intergenic variant |
| rs12434646 | 14 | T | C | 0.422 | $1.58 \times 10^{-08}$ | 0.970 [0.960, 0.979] | ???- | AREL1 | Intron variant |
| rs34306440 | 16 | A | G | 0.804 | $3.21 \times 10^{-10}$ | 1.028 [1.018, 1.039] | ++++ | CLEC16A | Intron variant |
| rs7196726 | 16 | A | G | 0.413 | $6.60 \times 10^{-79}$ | 1.068 [1.060, 1.077] | ++++ | ZNF646 | Missense variant |
| rs1421085 | 16 | T | C | 0.616 | $5.75 \times 10^{-26}$ | 0.963 [0.955, 0.970] | ---- | FTO | Intron variant |
| rs62066119 | 17 | T | C | 0.189 | $3.05 \times 10^{-13}$ | 1.040 [1.030, 1.050] | ++?+ | LINC02210 | Upstream gene variant |
| rs244304 | 17 | T | C | 0.708 | $3.65 \times 10^{-08}$ | 0.978 [0.971, 0.986] | ---- | STXBP4 | 3 prime UTR variant |
| rs61730306 | 18 | A | G | 0.913 | $3.27 \times 10^{-10}$ | 0.963 [0.952, 0.974] | ---- | DSG1 | Missense variant |
| rs2043332 | 19 | A | C | 0.292 | $2.00 \times 10^{-08}$ | 0.978 [0.971, 0.986] | ---- | DNM2 | Intron variant |
| rs438735 | 19 | A | C | 0.094 | $1.30 \times 10^{-14}$ | 0.955 [0.944, 0.966] | ---- | ARRDC2 | Downstream gene variant |
| rs2236506 | 20 | A | G | 0.785 | $8.96 \times 10^{-15}$ | 0.967 [0.959, 0.974] | ---- | RTEL1 RTEL1-TNFRSF6B | Missense variant |

Association analyses were conducted using logistic regression under an additive genetic model.
A two-sided Wald test was used to evaluate the null hypothesis that the effect size (beta) equals zero. Lead variants (SNP), their odds ratios and confidence intervals, p-values (P), allele frequencies for effect allele (AF), nearest genes, variant types (most severe consequence) and direction of the effect in each cohort (in the following order: UKB, FinnGen, EstBB, MVP; question marks indicating missingness of the SNP in a particular data) are presented.

gtexportal.org/home/) (Supplementary Data Tables S2 and S3). We identified a strong shared signal between *SLURP2* (rs10094888) and a structurally and functionally highly similar Ly6SF-group gene *LYNX1* expression in skin tissue and in dermatophytosis with posterior probabilities of 0.999 (*LYNX1*) and 0.987 (*SLURP2*). The results indicate the same causal variant for differential expression of *SLURP2* and *LYNX1* in dermatophytosis and skin tissue (Fig. 3A, B). These proteins are secreted primarily in the skin by keratinocytes and have been previously implicated in skin diseases[16].

We also discovered associations in canonical loci that have been earlier reported in infections and obesity, including the *FTO* locus. The lead variant at the *FTO* locus was the same variant that has been implicated as a causal variant in higher body mass index (rs1421085), and formal colocalization analysis showed a signal with *IRX3*, with a posterior probability of 0.86 (Fig. 3C), aligning with the signal reported earlier for obesity[17]. Although obesity-associated variants are located within the *FTO* gene, they regulate the expression of the distant *IRX3* gene through long-range chromatin interactions, explaining the colocalization signal at this gene region[17]. These findings may implicate the role of BMI (body mass index) in dermatophytosis, as suggested in previous studies[18].

In addition to showing the likely association of variants from *SLURP2* and *FTO* to dermatophytosis and aforementioned missense variants, our GWAS shows an association between the HLA region and dermatophytosis infections, aligning with previous studies[19,20]. The lead variant for the association (rs1794269, beta = 0.046 and $p = 1.42 \times 10^{-36}$) was located closest to the *HLA-DQB1* gene, but a correlation analysis suggests a strongest correlation with *DQB1*05:01* with $r = 0.48$. Formal HLA fine mapping in FinnGen supports the association of *DQB1*05:01* with dermatophytosis, with p-value of 0.005. The association between the *HLA* region and dermatophytosis highlights an overall immune signal in dermatophytosis.

### GWAS associations highlight the role of compromised keratin processing behind dermatophytosis

Dermatophytes are fungi with the unique ability to utilize keratin as a source of energy, allowing them to invade and colonize the outer layers of skin, hair, and nails, causing superficial infections[21]. We identified several lead variants that were located within or near genes involved in the keratin lifecycle, including the expression, differentiation, migration, and apoptosis of keratin-producing cells (keratinocytes). The strongest associations were identified in Profilaggrin (*FLG*)

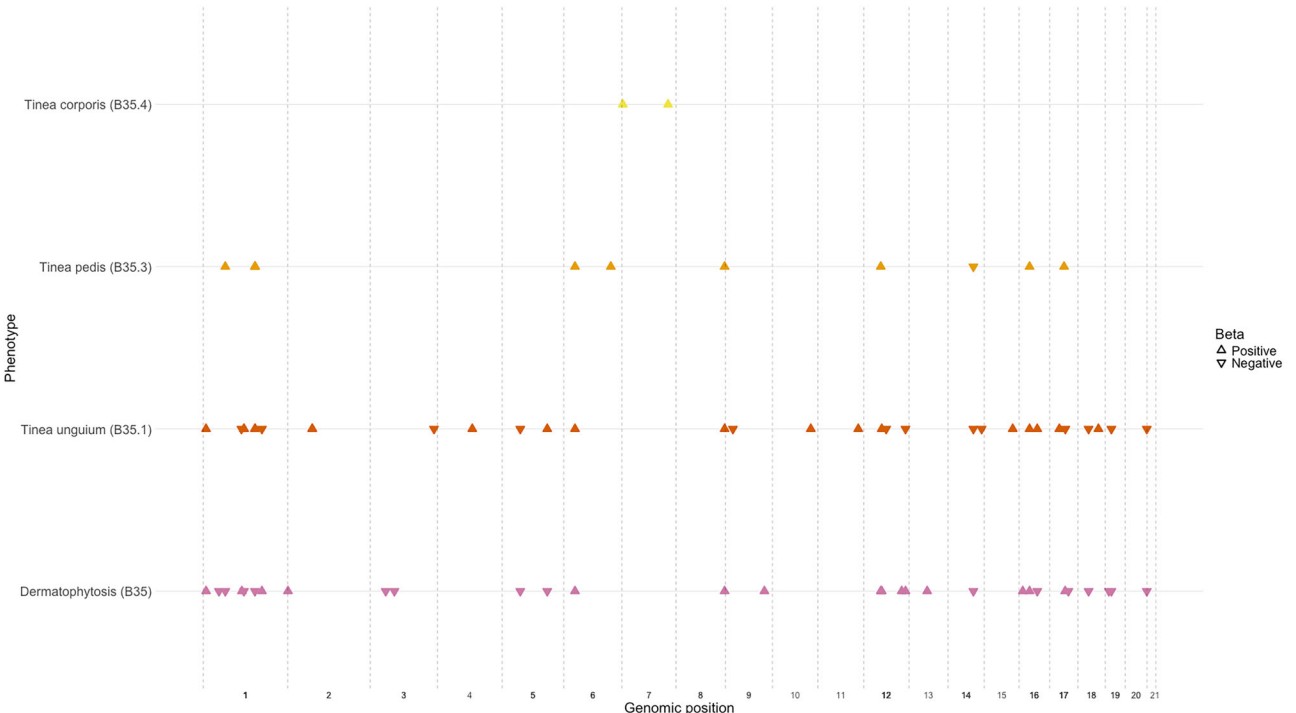

**Fig. 2 | Chromosomal position of the lead variants from the main dermatophytosis meta-analysis and subtype meta-analyses.** The direction of the effects is indicated with triangles. Supplementary Data are provided.

that forms the outermost layer of the skin, keratin 1 (*KRT1*), and *SLURP2*, all of which play critical roles in keratin function. These findings highlight keratin's crucial role in the development and progression of dermatophytosis. Many keratin-related proteins are essential for maintaining skin integrity and barrier function, which are key to protecting against fungal infections[22].

Moreover, we observed our strongest association at *ZNF646*. *ZNF646* belongs to a family of zinc finger proteins that act in transcriptional regulation and protein degradation[23]. This protein family is implicated in tissue development, particularly in the skin, where several family members can modulate keratinocyte gene expression and differentiation[23]. Zinc finger proteins have been earlier implicated in the regulation of *FLG*[23,24]. This observation raises a possibility that *ZNF646* modulates *FLG* expression. To test this, we examined the effect of the lead missense variant, rs7196726, at *ZNF646* on *FLG* and *KRT1* expression in the skin using the GTEx eQLT calculator (https://gtexportal.org/home/testyourown). We observed that *ZNF646* was a trans-eQTL for *FLG* expression in the sun-exposed skin (NES = −0.063 and $p = 0.0039$) as well as in the not sun-exposed skin (NES = −0.058 and $p = 0.031$), suggesting an effector role of *ZNF646* on *FLG*.

Another significant group of genes associated with dermatophytosis involves those related to immune defense, such as *HLA* genes, *IL13*, and *SH2B3* (*LNK*). These genes play crucial roles in antigen presentation, immune signaling, and cytokine production, all of which are essential for the body's defense against pathogens[25–27].

A third group of genes identified is related to obesity. Among these, the most notable was the association with *FTO* and *IRX3*, with *IRX3* mediating the functional effects of *FTO*[17]. The structure of the skin, the roles of each layer, and the potential causal genes for dermatophytosis—grouped by their function—are presented in Fig. 4.

**Variant annotations support associations with skin well-being, general immune defense and BMI**
To get a broader understanding of lead variants in dermatophytosis GWAS, we assessed their associations with other diseases and traits using FinnGen's annotation tool (https://anno.finngen.fi, https://

github.com/juhis/genetics-results-browser) covering all endpoints in FinnGen data freeze 12 (R12) (https://risteys.finregistry.fi/), FinnGen R12 and UKB combined meta-analysis and Open targets (https://platform.opentargets.org/). The strongest genome-wide significant ($p < 5 \times 10^{-8}$) signal for each variant is shown in Table 3. Besides the strongest associations presented in the table, most of the variants are associated with multiple or up to few hundred other traits at a genome-wide significant level.

Several lead variants are most strongly associated with circulating immune cells and autoimmune diseases, highlighting the connection of dermatophytosis to suboptimal internal immune defense. Secondly, we see associations with vitamin D levels, a vitamin that is essential for skin well-being and is known to regulate, for example, epidermal keratinocytes[28]. A few of the most significant genetic associations are also directly linked with skin diseases, such as atopic dermatitis or skin cancer. Lastly, we see a group of associations with traits related to high body weight.

**Stratified LDSC suggests association to skin and immune tissues**
Next, we performed a tissue-stratified linkage disequilibrium score regression (s-LDSC) analysis to identify which tissues are the most relevant for dermatophytosis infection (Fig. 5 and Supplementary Data Table S5). We show most significant associations with connective tissue ($p = 4.32 \times 10^{-10}$) and immune cells ($p = 1.96 \times 10^{-8}$). As skin tissue is included in the connective tissue, the most significant associations strengthen our other results regarding the relevance of skin integrity and well-being and the internal immune defense defending against dermatophytosis fungal infections.

To further study the specific related tissue types, we performed s-LDSC multi-tissue analysis with 80 immune cell types and chromatin marker combinations and 39 skin cell and chromatin marker combinations[29]. We identified the strongest association in the immune subset with T-helper cells (primary T helper 17 cells, PMA-I stimulated, H3K4me1, $p = 0.0005$) and in the skin subset with fibroblasts (foreskin fibroblast primary cells skin02, H3K27ac, $p = 0.005$) and keratinocytes (foreskin keratinocyte primary cells skin03, H3K4me3, $p = 0.024$). Only

**Table 2 | Missense lead variants and missense variants in LD with our lead variants reported with the effect gene, amino acid change, p-value, odds ratio and PolyPhen score describing the damaging effect of the amino acid change**

| CHR | GWAS lead variant | Effect allele | Non-effect allele | Missense variant in high LD | Gene | Amino acid change | LD | P | OR (95% CI) | PolyPhen score |
|---|---|---|---|---|---|---|---|---|---|---|
| 1 | rs2476601 | A | G | NA | PTPN22 | Trp620Arg | NA | $2.03 \times 10^{-08}$ | 1.032 [1.020, 1.044] | 0 |
| 1 | rs35273427 | T | C | NA | ADAM30 | Thr737Ala | NA | $1.40 \times 10^{-09}$ | 0.962 [0.951, 0.974] | 0 |
| 1 | rs558269137 | CACTG | C | NA | FLG FLG-AS1 | Ser761CysfsTer36 | NA | $1.96 \times 10^{-27}$ | 0.800 [0.768, 0.834] | NA |
| 5 | rs848* | A | C | rs20541 | IL13, TH2LCRR | Gln144Arg | $D' = 1$ $r^2 = 0.966$ | $1.28 \times 10^{-11}$ | 0.973 [0.965, 0.981] | 0 |
| 12 | rs1829637* | A | G | rs14024 | KRT1 | Lys633Arg | $D' = 0.996$ $r^2 = 0.987$ | $3.95 \times 10^{-15}$ | 0.970 [0.963, 0.977] | 0.0780 |
| | | | | rs10783528 | KRT77 | Asp336Asn | $D' = 1$ $r^2 = 0.888$ | $9.75 \times 10^{-12}$ | 1.025 [1.018, 1.032] | 0.0390 |
| 12 | rs3184504 | T | C | NA | SH2B3 | Trp262Arg | NA | $9.74 \times 10^{-10}$ | 1.022 [1.014, 1.030] | 0.0390 |
| 16 | rs7196726 | A | G | NA | ZNF646 | Gly1477Asp | NA | $6.60 \times 10^{-79}$ | 1.068 [1.059, 1.077] | 0.00300 |
| | | | | rs35713203 | ZNF646 | Gly921Ala | $D' = 0.983$ $r^2 = 0.936$ | $3.19 \times 10^{-76}$ | 1.068 [1.061, 1.076] | 0.0150 |
| | | | | rs7199949 | PRSS53 | Pro406Ala | $D' = 1$ $r^2 = 0.992$ | $1.90 \times 10^{-78}$ | 1.067 [1.060, 1.075] | 0 |
| | | | | rs9938550 | HSD3B7 | Thr250Ala | $D' = 0.911$ $r^2 = 0.801$ | $3.43 \times 10^{-57}$ | 1.058 [1.050, 1.065] | 0 |
| 18 | rs61730306 | A | G | NA | DSG1 | Lys537Arg | NA | $3.27 \times 10^{-10}$ | 0.963 [0.952, 0.974] | 0 |
| | | | | rs16961689 | DSG1 | Tyr528Ser | $D' = 1$ $r^2 = 1$ | $3.49 \times 10^{-10}$ | 0.963 [0.952, 0.974] | 0 |
| | | | | rs34302455 | DSG1 | Asp538Asn | $D' = 1$ $r^2 = 1$ | $3.84 \times 10^{-10}$ | 1.039 [1.027, 1.051] | 0 |
| 20 | rs2236506 | A | G | NA | RTEL1 RTEL1-TNFRSF6B | Ala758Ala | NA | $8.96 \times 10^{-15}$ | 0.967 [0.959, 0.974] | 0.00700 |

In the GWAS lead variant column, the variants marked with an asterisk (*) are not missense variants but have missense variants in high LD ($D' > 0.9$).

the association with PMA-I stimulated primary T helper cells (H3K4me1) passes the Bonferroni corrected $p$-value threshold (threshold $p$-value 0.0006 for immune cells and 0.001 for skin cells) (Supplementary Data Table S5).

### Dermatophytosis shows genetic correlation with other diseases of skin and high BMI

We employed LDSC to study the genetic correlation of dermatophytosis with other infections of the skin and subcutaneous tissue, certain infectious and parasitic diseases and obesity. By using our meta-analysis summary statistics and all relevant endpoints in FinnGen, we found associations with several skin disease endpoints (Fig. 6 and Supplementary Data Table S7), many of which include itchiness, rash or redness of the skin. The most significant associations were with erysipelas (ICD10: A46, $p = 3.5 \times 10^{-20}$), other disorders of skin and subcutaneous tissue (ICD10: L98, $p = 7.4 \times 10^{-16}$) and infections of skin and subcutaneous tissue (ICD10: L00-L08, $p = 1.5 \times 10^{-10}$). Moreover, we observed shared genetic architecture between scabies, herpes simplex infection and other bacterial disease endpoints ($P < 0.0001$). The finding highlights the similar etiology behind skin-related diseases and proposes that the same genetic variants may increase our susceptibility to several types of skin diseases.

In addition, we studied the genetic correlation between dermatophytosis and BMI due to the association of several lead variants with changes in BMI and found a strong correlation with obesity ($p = 4.4 \times 10^{-49}$) and BMI IRN (inverse rank normalized) ($p = 3.2 \times 10^{-35}$).

## Discussion

Our meta-analysis of over 250,000 cases and 1,300,000 controls identified 30 genetic loci associated with dermatophytosis, nine of which were either missense variants or in high LD with a missense variant. Many of the loci were observed in the dermatophytosis sub-type meta-analyses, suggesting a shared host defense mechanism

against dermatophyte infections, irrespective of the site of manifestation. Several of the associated loci were linked to keratin processing and skin integrity, immune defense against pathogens, or environmental factors, such as high BMI. Additionally, we identified skin and immune cells as the most relevant tissue types for dermatophytosis infection using stratified LDSC analysis for different tissue types (narrow tissue dataset). Finally, a genetic correlation analysis with other skin and subcutaneous tissue infections, parasitic diseases, and obesity revealed shared genetic architecture, highlighting commonalities between dermatophytosis, other skin diseases and high BMI. Overall, these findings implicate the role of immune mechanisms, environmental contributions from high BMI and vitamin D biology and most notably skin integrity and its barrier role, keratin biology and keratin processing in dermatophytosis.

Our findings suggest that genetic variation in keratin-related genes, including those influencing keratinocyte function and skin well-being—such as KRT1, KRT77, FLG, SLURP2, and ZNF646—play a critical role in susceptibility to dermatophyte infections. Since dermatophytes specifically target keratin using it as an energy source, it is noteworthy that genetic variation in these loci potentially can modulate disease susceptibility[21]. The findings highlight the role of keratin and barrier organs like the skin as the first line of defense against pathogens, such as fungi. Earlier genome-wide association studies have explored dermatophytosis and suggest one risk variant at the TINAG locus[30], but this finding was not replicated in the meta-analysis.

We identified associations between dermatophytosis and genetic variation in several key keratin-related genes, including KRT1, KRT77, and FLG. KRT1 encodes for keratin 1 protein, which is essential in maintaining the structural integrity of the skin by forming the cytoskeleton of keratinocytes, contributing to the skin's protective barrier against pathogens and environmental damage[31]. Moreover, keratin 1 is part of the nutrient source for the fungal dermatophytes. Missense variation in KRT1 may have two mechanisms that affect disease

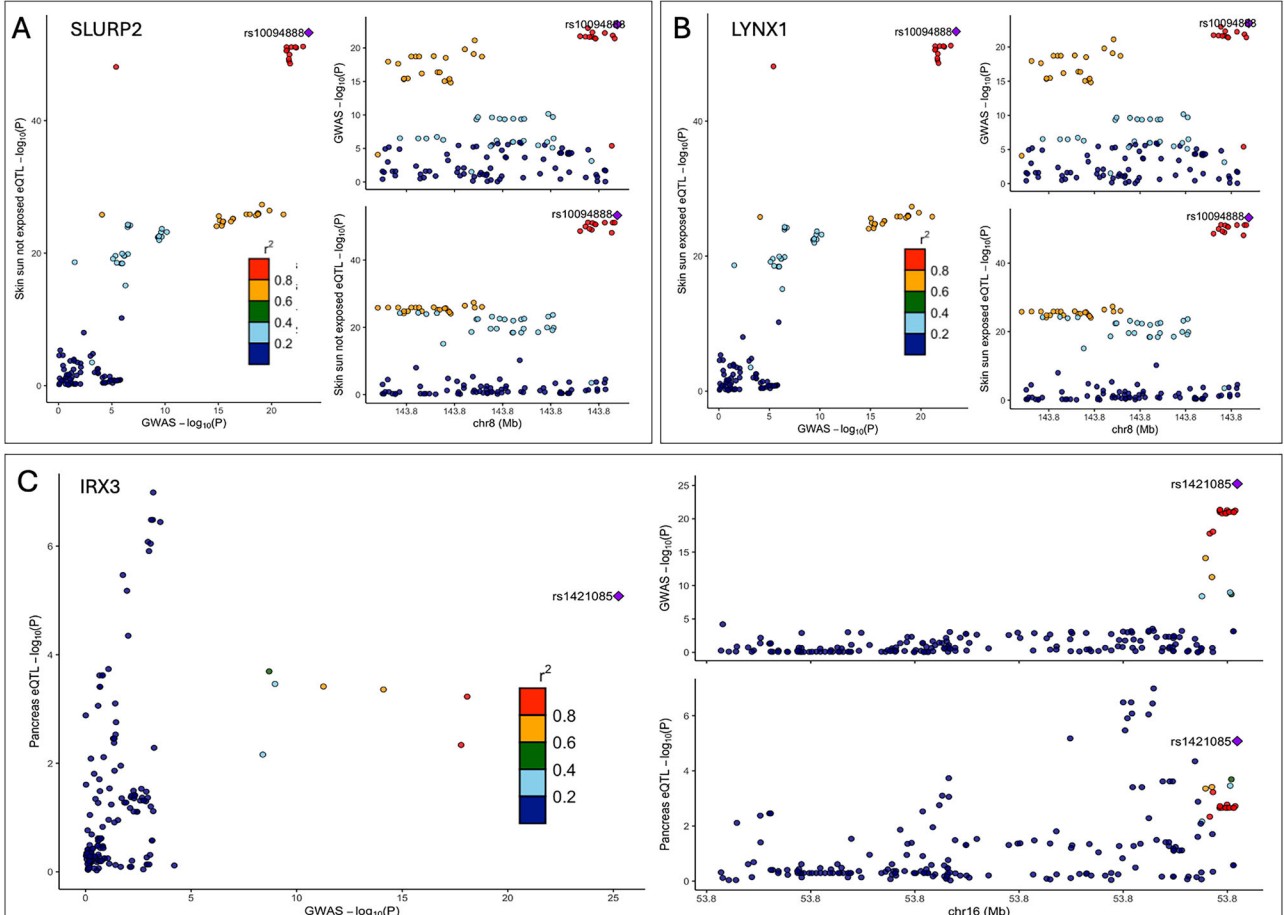

**Fig. 3 | Colocalization plots for dermatophytosis infection and relevant tissue.**
**A** Colocalization plot for SLURP2 and not sun-exposed skin tissue. **B** Colocalization plot for LYNX1 and sun-exposed skin tissue. **C** Colocalization plot for IRX3 and pancreatic tissue. eQTL *p*-values were obtained from the GTEx v.8 and are based on linear regression under an additive genetic model. GWAS *p*-values were calculated using REGENIE logistic regression under an additive model. All tests were two-sided. Plots illustrate $-\log_{10}(p\text{-value})$ of *SLURP2/LYNX1/IRX3* variants in dermatophytosis (x-axis) and their expression in the selected tissue (y-axis), regional associations of these variants with dermatophytosis (top right) and their regional association with RNA expression measured in the skin tissue in GTEx (bottom right).

susceptibility. It is possible that the protective missense variant may influence the structure and stability of the keratin filaments, improving the barrier function. Alternatively, the missense variant can affect the protein structure, making it harder to be degraded by the fungi. It should be also noted that the association with a missense variant does not necessarily mean that they would be the causal variant and instead could indicate LD with a regulatory variant. Therefore, the role of the missense variants should be addressed in future functional studies.

*FLG* is also directly involved in keratin processing, as it participates in the terminal differentiation of keratinocytes, where it aggregates keratin filaments into dense bundles, forming a protective layer of dead keratinocytes on the outer surface of the skin. This layer, called *stratum corneum*, provides a tough and protective barrier both against mechanical stress and pathogens[32]. Moreover, filaggrin (*FLG*) is crucial in skin hydration, as its breakdown products serve as natural moisturizing factors, retaining moisture in the skin and preventing dryness[33]. Additionally, these breakdown products help maintain a slightly acidic skin pH, inhibiting the growth of pathogenic microbiota[34]. Mutations in *KRT1* and *FLG* have been linked to skin disorders, such as epidermolytic hyperkeratosis[35], atopic dermatitis[33], and ichthyosis vulgaris[34].

Furthermore, colocalization analysis suggests the same genetic variant behind dermatophytosis and differential expression of *SLURP2* and *LYNX1* in the skin tissue. Both genes are expressed by keratinocytes and may play roles in skin homeostasis and the pathogenesis of skin

disorders by modulating nicotinic acetylcholine receptor functions[16]. Notably, *SLURP2* has been shown to promote keratinocyte hyperproliferation by inhibiting apoptosis. It also connects with the adaptive immune system and affects T-cell differentiation and activation[36]. Similar to *KRT1* and *FLG*, *SLURP2* has been linked to skin diseases, particularly Mal de Meleda, a rare genetic disorder causing thickened skin on the palms and soles[37].

Lastly, linked to skin well-being, our strongest association with dermatophytosis is with a missense variant in the *ZNF646* gene. *ZNF646* belongs to a family of zinc finger proteins that act in transcriptional regulation and protein degradation. This protein family is implicated in tissue development, particularly in the skin, where several family members can modulate keratinocyte gene expression and differentiation[23]. Our findings suggest a regulatory effect of the *ZNF646* missense variant on *FLG* expression, but how *ZNF646* participates in dermatophytosis needs to be evaluated in future functional studies.

In addition to identifying genes linked directly to keratin processing and skin health, we showed a second group of lead variants associated with immune defense. The strongest association was found within the HLA region (*HLA-DQB1*), which plays a critical role in defending against infections by presenting pathogen antigens to T-cells[25]. This finding highlights the involvement of canonical immune and autoimmune mechanisms in dermatophytosis. While dermatophytosis is primarily an infection, the variants that were associated with it were also associated with autoimmune traits. The overall

**Fig. 4 | Dermatophytosis-associated genes and their roles.** Majority of dermatophytosis-associated genes were linked with keratin processing and skin integrity, systemic immune defense against pathogens or obesity. Figure created in BioRender. Haapaniemi, H. (2025) https://BioRender.com/xe97y42.

connection between infectious and immune loci that associate with autoimmunity needs to be further explored in future studies.

Other notable immune-related associations were with *IL13* and *SH2B3*. Interleukin-13 (*IL13*) is involved in allergy, inflammation, and immune defense against parasites, such as helminths[26]. Interestingly, *IL13* also influences skin integrity by downregulating filaggrin and involucrin—two key components of the skin barrier. It also drives the release of itch-inducing molecules (pruritogens) and inflammatory cytokines, exacerbating chronic itch, a common symptom of dermatophytosis. *IL13* has been previously linked to atopic dermatitis[38]. *SH2B3* (*LNK*), on the other hand, plays a central role in modulating immune cell signaling by interacting with cytokine pathways and T/B cell receptor signaling, with associations to multiple autoimmune diseases[27].

A third group of genes identified in dermatophytosis GWAS were related to obesity, with the most notable being *FTO* and *IRX3*. *FTO* locus is a well-known genetic region responsible for energy homeostasis and body weight. Variation in its intronic region are among the strongest associations with obesity[17]. The functional effects of *FTO* variants are mediated through homeobox genes, including *IRX3* and the variants at the *FTO* locus act as its long-range enhancers[17]. Additionally, variant annotations relate our missense variant in *ZNF646* with high BMI and obesity, further supporting the connection between metabolic health and susceptibility to dermatophytosis.

Our findings underscore the critical role of keratin processing and skin well-being, immune defense, and BMI-related genes in dermatophytosis susceptibility. Variant annotations further support these three categories. Tissue-type genetic correlation analysis identified immune cells and connective tissue as the most relevant tissue types in dermatophytosis infection. We also demonstrated significant genetic correlations between dermatophytosis and several other conditions, including skin and subcutaneous infections (such as other disorders of the skin and subcutaneous tissue,

psoriasis and urticaria), certain infectious and parasitic diseases (erysipelas), and obesity (BMI).

Despite the substantial scale of our GWAS on dermatophytosis, several factors should be considered when interpreting our findings. Given that multiple skin infections can present with similar symptoms, it is possible that we also capture variants that are not specific to dermatophytosis but implicate shared or overlapping biology with other diseases that share similar symptomatology. Second, although we increased sample size and power by utilizing data from multiple cohorts, our cases and controls are predominantly of European ancestry. While the MVP dataset includes individuals from other ancestries, it is likely that we do not capture the full genetic architecture of dermatophytosis. Lastly, we observed that some lead variants were not consistently present across all four datasets, likely due to differences in genotyping arrays and imputation platforms, which may reduce the power to detect significant associations.

This study highlights the complex interplay between genetic factors involved in keratin production, immune response, and metabolic regulation in determining susceptibility to dermatophytosis. By identifying key genetic variants across multiple biobanks, we provide insights into the biological mechanisms underlying dermatophyte infection and shed light on potential targets for prevention and treatment. Our results underscore the importance of skin barrier integrity, immune defenses, and metabolic health in protecting against fungal infections. These findings pave the way for future research into precision medicine approaches for dermatophytosis, aiming to develop personalized strategies for individuals at higher genetic risk of infection.

## Methods
### Cohorts
FinnGen is a population-based public-private population cohort established in 2017[39]. The study combines genetic data with electronic

**Table 3 | Most significant associations for our dermatophytosis meta-analysis GWAS lead variants with endpoints from FinnGen release 12 (R12), FinnGen R12 and UKB meta-analysis and Open targets**

| Variant | RSID | Effect allele | Non-effect allele | Nearest gene | Most significant association | *p*-value | Beta |
|---|---|---|---|---|---|---|---|
| 1-113834946-A-G | rs2476601 | A | G | PTPN22 | Autoimmune diseases | $10.00 \times 10^{-403}$ | −0.25 |
| 1-119894128-T-C | rs35273427 | T | C | ADAM30 | Type 2 diabetes wide definition | $6.31 \times 10^{-14}$ | 0.09 |
| 1-152312600-CACTG-C | rs558269137 | CACTG | C | FLG | Vitamin D levels | $3.70 \times 10^{-97}$ | 0.15 |
| 1-172877652-A-G | rs10912488 | A | G | RP1-15D23.2 | Crohn's disease | $1.87 \times 10^{-21}$ | 0.17 |
| 1-65014876-A-G | rs506025 | A | G | JAK1 | Eosinophil counts | $4.40 \times 10^{-30}$ | 0.04 |
| 1-8545608-A-T | rs6577497 | A | T | RERE | Heel bone mineral density | $4.60 \times 10^{-28}$ | −0.02 |
| 2-623863-GTTC-G | rs60383093 | G | GTTC | THEM18 | Body-mass index inverse-rank normalized | $1.73 \times 10^{-101}$ | 0.06 |
| 20-63690302-G-A | rs2236506 | A | G | RTEL1-TNFRSF6B | Atopic dermatitis | $1.00 \times 10^{-28}$ | 0.10 |
| 3-45922676-T-C | rs11130078 | T | C | FYCO1 | Monocyte percentage of white cells | $1.20 \times 10^{-25}$ | 0.02 |
| 3-71433933-G-A | rs62246017 | A | G | FOXP1 | Malignant neoplasm of skin, excluding all cancers (controls excluding all cancers) | $2.00 \times 10^{-31}$ | 0.08 |
| 5-132660808-A-C | rs848 | A | C | IL13 | Eosinophil counts | $7.48 \times 10^{-126}$ | −0.06 |
| 5-53277913-T-A | rs37807 | A | T | CTD-2143A15.1 | Corneal resistance factor (left) | $5.67 \times 10^{-22}$ | −0.14 |
| 6-32706117-C-T | rs1794269 | T | C | MTCO3P1 | Type 1 diabetes definitions combined | $10.00 \times 10^{-656}$ | 0.87 |
| 8-142762416-C-T | rs10094888 | T | C | SLURP2. LYNX1 | Total testosterone levels | $1.40 \times 10^{-16}$ | −0.02 |
| 12-111446804-T-C | rs3184504 | T | C | SH2B3 | Eosinophil counts | $2.23 \times 10^{-308}$ | −0.10 |
| 12-122700003-T-C | rs612057 | T | C | RP11-324E6.6 | Plateletcrit | $2.30 \times 10^{-11}$ | 0.01 |
| 12-52695529-G-A | rs1829637 | A | G | KRT77 | Basal cell carcinoma, excluding all cancers (controls excluding all cancers) | $2.51 \times 10^{-08}$ | −0.04 |
| 13-53167993-A-G | rs1885767 | A | G | LINC01065 | Sleeplessness/insomnia | $2.00 \times 10^{-12}$ | −0.01 |
| 14-74711990-C-T | rs12434646 | T | C | AREL1 | Appendicular lean mass | $8.42 \times 10^{-18}$ | −0.02 |
| 16-11121178-A-G | rs34306440 | A | G | CLEC16A | Eosinophil counts | $1.62 \times 10^{-105}$ | −0.05 |
| 16-31080754-G-A | rs7196726 | A | G | ZNF646 | Serum 25-hydroxyvitamin D levels | $2.23 \times 10^{-308}$ | −0.01 |
| 16-53767042-T-C | rs1421085 | T | C | FTO | Body-mass index inverse-rank normalized | $3.48 \times 10^{-319}$ | 0.08 |
| 17-45617831-C-T | rs62066119 | T | C | LINC02210-CRHR1 | Mean spheric corpuscular volume | $9.00 \times 10^{-128}$ | −0.06 |
| 18-31339948-A-G | rs61730306 | A | G | DSG1 | Serum 25-hydroxyvitamin D levels | $2.23 \times 10^{-308}$ | −0.02 |
| 19-10780563-C-A | rs2043332 | A | C | DNM2 | Statin medication | $1.94 \times 10^{-47}$ | −0.08 |
| 19-18015397-C-A | rs438735 | A | C | ARRDC2 | Monocyte count | $3.30 \times 10^{-19}$ | −0.03 |
| 20-63690302-G-A | rs2236506 | A | G | RTEL1-TNFRSF6B | Atopic dermatitis | $1.00 \times 10^{-28}$ | 0.10 |

*P*-values and betas are extracted from GWAS summary statistics for the top trait associated with each lead variant identified in the dermatophytosis analysis.

health record data, including International Classification of Diseases (ICD) codes spanning an individual's entire lifespan, derived from primary care registers, hospital inpatient and outpatient visits and drug prescriptions of 520,000 participants. The project aims to improve understanding of the genetic etiology of diseases and disorders, potentially leading to drug development.

The UKB is a prospective open-access study containing over 500,000 individuals aged 40–69 years upon entry to the study between 2006 and 1010[40]. At the time of the entry to the cohort, a variety of health and lifestyle measures were collected, and blood and urine samples were taken for genetic and biochemistry analysis. Hospital inpatient (HES; N ~ 470,000) and primary care (GP; N ~ 231,000) records were later linked up to provide longitudinal data on disease diagnosis, operations, deaths, medications, and deaths. In this study, we used publicly available summary statistics from the pan-UKB study[41] for main dermatophytosis and self-analyzed summary statistics for subtypes.

The EstBB is a population-based biobank with 212,955 participants[42]. Information on ICD-10 codes is obtained through regular linking with the National Health Insurance Fund and other relevant databases. The majority of the electronic health records have been collected since 2004.

The Million Veteran Project (MVP) is a longitudinal cohort study of diverse U.S. Veterans looking at how genes, lifestyle, military

experiences, and exposures affect health and wellness[43]. It combines genetic data with electronic health records of 635,969 participants (data freeze 4) across four ethnic groups. In this study, we extracted data from the publicly released summary statistics for MVP for dermatophytosis for all ancestries (AFR, AMR, EAS, and EUR).

### Ethics statements

Study subjects in FinnGen provided informed consent for biobank research, based on the Finnish Biobank Act. Alternatively, separate research cohorts, collected prior the Finnish Biobank Act came into effect (in September 2013) and the start of FinnGen (August 2017), were collected based on study-specific consents and later transferred to the Finnish biobanks after approval by Fimea (Finnish Medicines Agency), the National Supervisory Authority for Welfare and Health. Recruitment protocols followed the biobank protocols approved by Fimea. The Coordinating Ethics Committee of the Hospital District of Helsinki and Uusimaa (HUS) statement number for the FinnGen study is Nr HUS/990/2017.

The FinnGen study is approved by Finnish Institute for Health and Welfare (permit numbers: THL/2031/6.02.00/2017, THL/1101/5.05.00/2017, THL/341/6.02.00/2018, THL/2222/6.02.00/2018, THL/283/6.02.00/2019, THL/1721/5.05.00/2019, and THL/1524/5.05.00/2020), digital and population data service agency (permit numbers:

VRK43431/2017-3, VRK/6909/2018-3, and VRK/4415/2019-3), the Social Insurance Institution (permit numbers: KELA 58/522/2017, KELA 131/522/2018, KELA 70/522/2019, KELA 98/522/2019, KELA 134/522/2019, KELA 138/522/2019, KELA 2/522/2020, and KELA 16/522/2020), findata permit numbers THL/2364/14.02/2020, THL/4055/14.06.00/2020, THL/3433/14.06.00/2020, THL/4432/14.06/2020, THL/5189/14.06/2020, THL/5894/14.06.00/2020, THL/6619/14.06.00/2020, THL/209/14.06.00/2021, THL/688/14.06.00/2021, THL/1284/14.06.00/2021, THL/1965/14.06.00/2021, THL/5546/14.02.00/2020, THL/2658/14.06.00/2021, and THL/4235/14.06.00/2021, statistics finland (permit numbers: TK-53-1041-17 and TK/143/07.03.00/2020 (earlier TK-53-90-

20) TK/1735/07.03.00/2021, TK/3112/07.03.00/2021) and Finnish Registry for Kidney Diseases permission/extract from the meeting minutes on 4th July 2019.

The Biobank Access Decisions for FinnGen samples and data utilized in FinnGen Data Freeze 11 include: THL Biobank BB2017_55, BB2017_111, BB2018_19, BB_2018_34, BB_2018_67, BB2018_71, BB2019_7, BB2019_8, BB2019_26, BB2020_1, BB2021_65, Finnish Red Cross Blood Service Biobank 7.12.2017, Helsinki Biobank HUS/359/2017, HUS/248/2020, HUS/430/2021 §28, §29, HUS/150/2022 §12, §13, §14, §15, §16, §17, §18, §23, §58, §59, HUS/128/2023 §18, Auria Biobank AB17-5154 and amendment #1 (August 17 2020) and amendments BB_2021-0140, BB_2021-0156 (August 26 2021, Feb 2 2022), BB_2021-0169, BB_2021-0179, BB_2021-0161, AB20-5926 and amendment #1 (April 23 2020) and it´s modifications (Sep 22 2021), BB_2022-0262, BB_2022-0256, Biobank Borealis of Northern Finland_2017_1013, 2021_5010, 2021_5010 Amendment, 2021_5018, 2021_5018 Amendment, 2021_5015, 2021_5015 Amendment, 2021_5015 Amendment_2, 2021_5023, 2021_5023 Amendment, 2021_5023 Amendment_2, 2021_5017, 2021_5017 Amendment, 2022_6001, 2022_6001 Amendment, 2022_6006 Amendment, 2022_6006 Amendment, 2022_6006 Amendment_2, BB22-0067, 2022_0262, 2022_0262 Amendment, Biobank of Eastern Finland 1186/2018 and amendment 22§/2020, 53§/2021, 13§/2022, 14§/2022, 15§/2022, 27§/2022, 28§/2022, 29§/2022, 33§/2022, 35§/2022, 36§/2022, 37§/2022, 39§/2022, 7§/2023, 32§/2023, 33§/2023, 34§/2023, 35§/2023, 36§/2023, 37§/2023, 38§/2023, 39§/2023, 40§/2023, 41§/2023, Finnish Clinical Biobank Tampere MH0004 and amendments (21.02.2020 & 06.10.2020), BB2021-0140 8§/2021, 9§/2021, §9/2022, §10/2022, §12/2022, 13§/2022, §20/2022, §21/2022, §22/2022, §23/2022, 28§/2022, 29§/2022, 30§/2022, 31§/2022, 32§/2022, 38§/2022, 40§/2022, 42§/2022, 1§/2023, Central Finland Biobank 1-2017, BB_2021-0161, BB_2021-0169, BB_2021-0179, BB_2021-0170, BB_2022-0256, BB_2022-0262, BB22-0067, Decision allowing to continue data processing until 31st Aug 2024 for projects: BB_2021-0179, BB22-0067, BB_2022-0262, BB_2021-0170, BB_2021-0164, BB_2021-0161, and BB_2021-0169, and Terveystalo Biobank STB 2018001 and amendment 25th Aug 2020, Finnish Hematological Registry and Clinical Biobank decision 18th June 2021, Arctic biobank P0844: ARC_2021_1001.

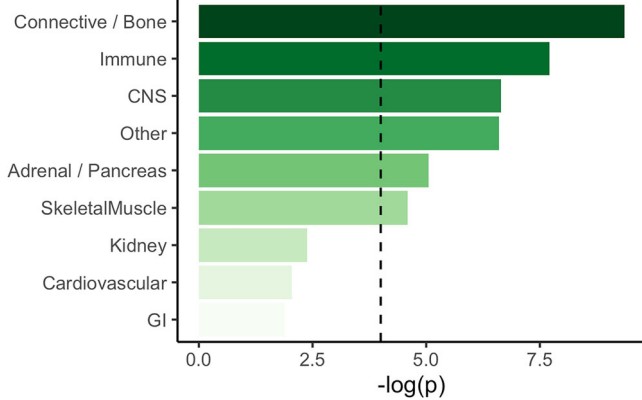

**Fig. 5 | Stratified LDSC (S-LDSC) analysis in dermatophytosis.** Connective tissue and immune cells are identified as the most relevant tissue types for dermatophytosis infection. S-LDSC tests whether functional annotations explain more heritability than expected by chance. *P*-values were derived from a two-sided Z-test of the null hypothesis that the regression coefficient for each annotation equals zero. The x-axis shows $-\log_{10}(p\text{-values})$ for enrichment of SNP heritability in each tissue-specific annotation. The dashed line indicates the Bonferroni-corrected significance threshold for association. Central nervous system (CNS), gastro-intestinal (GI).

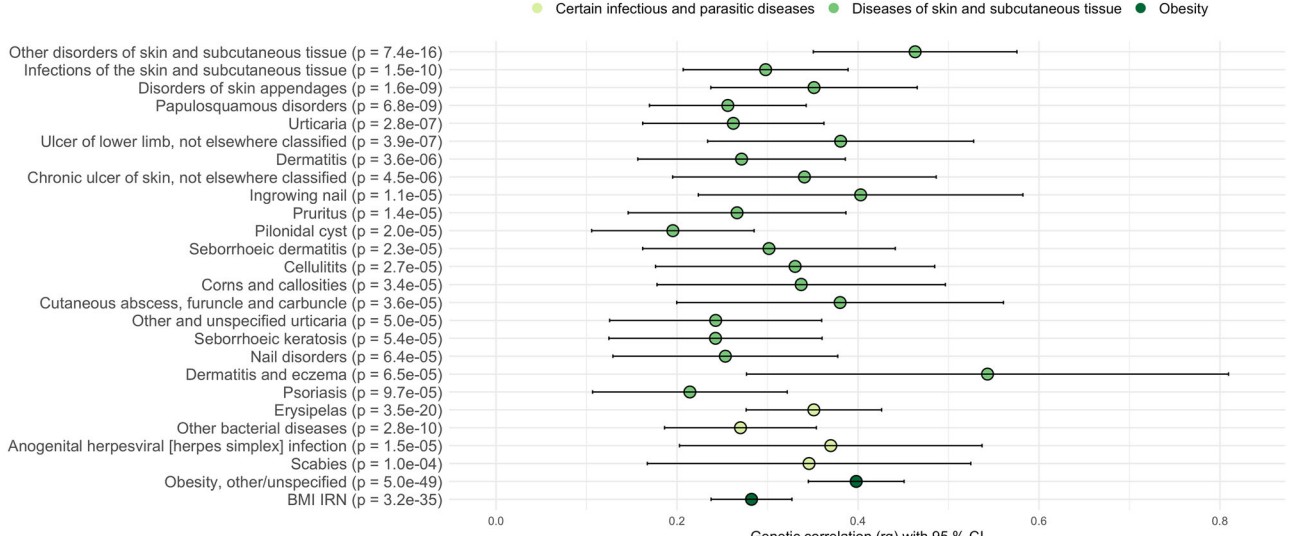

**Fig. 6 | Genetic correlation between dermatophytosis and FinnGen endpoints of other infections of skin and subcutaneous tissue, certain infectious and parasitic diseases and obesity.** Statistically significant results are presented as mean genetic correlation estimates (rg), with error bars representing 95% confidence intervals (mean ± 1.96 × standard error). Genetic correlations and *p*-values were calculated using LDSC software. The statistical test was a two-sided Z-test evaluating the null hypothesis that rg = 0. *P*-values are shown as exact values with no multiple testing correction. Each genetic correlation estimate represents a pairwise comparison between GWAS summary statistics from dermatophytosis and another trait. The case and control numbers of the other infections of skin and subcutaneous tissue, certain infectious and parasitic diseases and obesity are listed in the Supplementary Data S7.

The activities of the EstBB are regulated by the Human Genes Research Act, which was adopted in 2000 specifically for the operations of the EstBB. Individual-level data analysis in the EstBB was carried out under ethical approval 1.1-12/624 from the Estonian Committee on Bioethics and Human Research (Estonian Ministry of Social Affairs), using data according to release application 6-7/GI/33543 from the EstBB.

The North West multi-center research ethics committee has granted the research tissue bank approval for the UKB that covers the collection and distribution of data and samples (http://www.ukbiobank.ac.uk/ethics/). Our work was performed under the UKB application numbers 22627 (main meta-analysis) and 6818 (subtype analysis). All participants included in the conducted analyses have given a written consent to participate.

We would like to thank the participants and researchers from the UKB who contributed or collected data.

No compensation was given to the participants.

## Genotyping and quality control

FinnGen R12 contains genetic data for 520,210 individuals. The samples were genotyped using Illumina (Illumina) and Affymetrix arrays (Thermo Fisher Scientific). The array consisted of 735,145 probes looking for 655,973 variants consisting of core backbone variants for imputation, rare coding variants enriched in the Finnish population, variants for KIR and HLA haplotypes, disease-specific markers and pharmacogenomic markers. Genotyping data produced with previous chip platforms and reference genome builds were lifted over to build v.38 (GRCh38/hg38). For sample-wise quality control, individuals exhibiting a discrepancy between genetically inferred sex and reported sex in registries, high genotype missingness (>5%), and excess heterozygosity (±4 standard deviations) were excluded. For variant-level QC, variants with high missingness (>2%), low Hardy–Weinberg equilibrium ($P < 1 \times 10^{-6}$), and a minor allele count < 3 were filtered out. Chip-genotyped samples were pre-phased with Eagle 2.3.5 and imputed with the Finnish-specific SISu v4 imputation reference panel. Post-imputation quality control involved excluding variants with INFO score <0.7[39].

All EstBB participants have been genotyped at the core genotyping lab of the Institute of Genomics, University of Tartu, using Illumina Global Screening Array v3.0_EST. Samples were genotyped, and PLINK format files were created using Illumina GenomeStudio v2.0.4. Individuals were excluded from the analysis if their call rate was <95%, if they were outliers of the absolute value of heterozygosity (>3 SD from the mean) or if the sex defined based on heterozygosity of the X chromosome did not match the sex in phenotype data[44]. Before imputation, variants were filtered by call rate <95%, HWE $p < 1 \times 10^{-4}$ (autosomal variants only), and minor allele frequency <1%. Genotyped variant positions were in build 37 and were lifted over to build 38 using Picard. Phasing was performed using the Beagle v5.4 software[45]. Imputation was performed with Beagle v5.4 software (beagle.22Jul22.46e.jar) and default settings. The dataset was split into batches of 5000. A population-specific reference panel consisting of 2695 WGS samples was utilized for imputation, and standard Beagle hg38 recombination maps were used. Based on the principal component analysis, samples which were not from European ancestry individuals were removed. Duplicate and monozygous twin detection was performed with KING 2.2.7[46], and one sample was removed from the pair of duplicates.

A total of 488,477 participants in the UKB were genotyped, including 49,950 individuals using the Affymetrix (Thermo Fisher Scientific) UK BiLEVE Axiom Array and 438,427 using the closely related Affymetrix UKB Axiom Array. These platforms captured up to 825,927 SNPs and short indels, prioritizing known coding variants, previously disease-associated variants, and ancestry-informative markers to ensure robust imputation performance. DNA was extracted from blood samples collected during baseline assessments conducted

between 2006 and 2010. Genotyping was performed in 106 sequential batches, resulting in genotype calls for 812,428 unique variants across 489,212 participants. For genome-wide association study (GWAS) analyses, genotype data were filtered using PLINK2 (v2.0.0, 24 Nov 2024) to remove variants with call rates below 99%, minor allele frequencies under 1%, Hardy-Weinberg equilibrium exact test $p$-values less than 1e-15, and variants with linkage disequilibrium ($r^2 > 0.9$) within 100 kb windows. Genotype and sequencing data were imputed using the TOPMed R2 reference panel by the TOPMed Informatics Research Center.

## GWAS

Main dermatophytosis (B35) GWAS in FinnGen was conducted using the REGENIE v3.3 pipeline (https://github.com/FINNGEN/regenie-pipelines), adjusting for age, sex, chip, batch and first ten principal components[47].

Association analysis in the EstBB was carried out for all variants with an INFO score > 0.4 using the additive model as implemented in REGENIE v3.0.3 with standard binary trait settings[47]. Logistic regression was carried out with adjustment for current age, age², sex and 10 PCs as covariates, analyzing only variants with a minimum minor allele count of 2.

UKB summary statistics were readily available from the pan-UKB study (only European subset)[41] and MVP summary statistics from their web server (https://phenomics.va.ornl.gov/pheweb/gia/meta/pheno/Phe_110).

In all the cohorts, sex has been genetically determined. All studies have been adjusted with sex. The case and control definitions, numbers of cases and controls, used GWAS software and used covariates in each GWAS analysis are presented in Table 4.

In addition to the main dermatophytosis GWAS analyses, we performed subtype-specific GWAS analyses in FinnGen, EstBB and UKB using REGENIE. The analyses were adjusted with ten first principal components, age, sex and genotyping array. In each subtype-specific analysis, all dermatophytosis cases were excluded from the controls. MVP summary statistics for three of the phenotypes were obtained from their web server (https://phenomics.va.ornl.gov/pheweb/gia/meta/). Case and control numbers can be found from Supplementary Data S8.

## Meta-analysis

We conducted a meta-analysis with summary statistics from FinnGen, EstBB, UKB, and MVP for the main dermatophytosis and each subtype using the standard error method in METAL software[48]. The variants are annotated according to genome build 38.

The Manhattan plots for meta-analyses were plotted using R version 4.3.1 (packages: qqman and RColorBrewer).

## HLA fine mapping

Association testing for HLA variants in the FinnGen cohort was conducted using multivariate logistic regression to elucidate the relationships between individual HLA alleles and dermatophytosis infection. Multivariate logistic regression analysis was adjusted for age at death or the end of follow-up, sex, and the first 10 genetic principal components accounting for population structure. Multivariate logistic regression was performed in a stepwise manner by sequentially adding the most strongly associated HLA allele as a covariate to the analysis. This iterative process was repeated until no significant ($p < 0.05$) alleles remained. The multivariate logistic regression analyses were conducted using R 4.0.1.

HLA imputation in FinnGen was performed for *HLA A, HLA B, HLA C, HLA DRB1, HLA DQA1, HLA DQB1, HLA DPB1, HLA DRB3, HLA DRB4*, and *HLA DRB5* using the R library HIBAG (HLA genotype imputation with attribute bagging), as described by Ritari et al.[49]. The HLA imputation of FinnGen data is based on a set of SNPs directly genotyped on

**Table 4 | Cohorts structure used for running the GWAS**

| Cohort | Case definition | Control exclusions | N (cases) | N (controls) | GWAS software | Covariates used |
|---|---|---|---|---|---|---|
| FinnGen | ICD10: B35 | ICD10: B35 | 27662 | 471729 | REGENIE[47] | Age, sex, first 10 PCs, genotyping batches |
| UKB | ICD10: B35 ICD9: 110 | ICD10: B35\|B36\|B37\|B38\|B39\|B40\|B41\|B42\| B43\|B44\|B45\|B46\|B47\|B48\|B49; ICD9: 110\|111\|112\|113\|114\|115\|116\|117\|118\|119 | 27755 | 380368 | SAIGE[59] | Age, sex, age * sex, age$^2$, age$^2$ * sex, first 10 PCs |
| EstBB | ICD10: B35 | ICD10: B35, B36, B37, B38, B39, B4 | 50241 | 106586 | REGENIE[47] | Age, sex, age$^2$, first 10 PCs and genotype batch control |
| MVP biobank | ICD9: 110 | ICD9: 110 | 151164 | 413818 | SAIGE[59] | Age, sex, and first 10 principal components |

Case and control definitions, numbers of cases and controls, used GWAS software, and used covariates in each GWAS analysis included in the meta-analysis are listed below.

the FinnGen array. A Finnish reference panel, genotyped at clinical grade accuracy (4-digits, amino acid), was used.

## Colocalization analysis

To assess the shared association of our lead variants to dermatophytosis and tissue-specific eQTLs, we performed colocalization analysis. For the analysis, we used meta-analysis summary statistics from dermatophytosis from a region ±50,000 base pairs around our lead variants and imported eQTL association statistics from GTEx[15] (https://gtexportal.org/home/) for the same region for all tissues.

Colocalization was performed using the R package Coloc (v5.1.0.1 in R v4.2.2)[50] and co-localization plots were generated with the LocusCompareR R package (v1.0.0)[51] using LD r$^2$ from 1000 Genomes[52] European-ancestry samples.

## Tissue and cell type-specific analyses

To study relevant tissue and cell types for dermatophytosis infection, we employed the stratified LDSC method[29]. First, we assessed relevant tissue types using cell-type groups data as used in Finucane et al. consisting of two gene expression datasets, GTEx project and "Franke lab" dataset with 205 tissues and cell types, that were classified into nine categories (connective/bone, immune, other, CNS, skeletal muscle, liver, adrenal/pancreas, kidney, GI, and cardiovascular)[53–55]. Second, we studied relevant cell types using the Multi_tissue_chromatin_1000Gv3_ldscores dataset from Finucane et al.[29] composed of chromatin data from Roadmap Epigenomics and ENCODE projects. This dataset contains 489 tissue-specific chromatin-based annotations from peaks for six epigenetic marks (H3K27ac, H3K4me1, H3K4me3, H3K9ac, H3K36me3, and DHS)[56,57]. We used only annotations related to skin and immune cells, resulting in 80 immune cell type and chromatin marker combinations and 39 skin cell and chromatin marker combinations.

## Genetic correlation

We performed genetic correlation between dermatophytosis infection and FinnGen endpoints of other infections of skin and subcutaneous tissue, certain infectious and parasitic diseases and obesity using the LD score regression method[58]. HapMap 3 SNP list and European LD score files, which are provided with the software, were used in our LD score regression analyses. For dermatophytosis, we used summary statistics from our meta-analysis, and for FinnGen, we used all endpoints in categories AB1 and L12, as well as selected BMI-related endpoints (BMI_IRN and E4_OBESITYNAS). Further information on these FinnGen endpoints can be found at https://risteys.finregistry.fi/. When assessing the relevance of the genetic correlation, we adjusted the p-value with Bonferroni correction. The forest plot for genetic correlation was generated using the ggplot2 package in R (version 4.4.1).

## Reporting summary

Further information on research design is available in the Nature Portfolio Reporting Summary linked to this article.

## Data availability

The summary statistics data of dermatophytosis meta-analysis generated in this study have been deposited in the GWAS catalog (https://www.ebi.ac.uk/gwas/) under accession codes GCST90809868-GCST90809877. The individual-level data from the biobanks is limited to authorized researchers due to privacy laws. The summary-level data created in this study is provided in the Supplementary Data file.

## Code availability

We have used REGENIE from https://github.com/FINNGEN, the metal package for meta-analysis of the summary statistics https://github.com/statgen/METAL, the COLOC package for colocalization analysis https://github.com/chr1swallace/coloc and the LDSC pipeline for running genetic correlation and partitioned heritability (tissue type analysis) https://github.com/bulik/ldsc. Plotting codes can be shared upon request from the corresponding author.

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

## Acknowledgements

We want to acknowledge the participants and investigators of the FinnGen study. The FinnGen project is funded by two grants from Business Finland (HUS 4685/31/2016 and UH 4386/31/2016) and the following industry partners: AbbVie Inc., AstraZeneca UK Ltd, Biogen MA Inc., Bristol Myers Squibb (and Celgene Corporation & Celgene International II Sàrl), Genentech Inc., Merck Sharp and Dohme LCC, Pfizer Inc., GlaxoSmithKline Intellectual Property Development Ltd., Sanofi US Services Inc., Maze Therapeutics Inc., Janssen Biotech Inc, Novartis Pharma AG, and Boehringer Ingelheim International GmbH. Following biobanks are acknowledged for delivering biobank samples to FinnGen: Auria Biobank (www.auria.fi/biopankki), THL Biobank (www.thl.fi/biobank), Helsinki Biobank (www.helsinginbiopankki.fi), Biobank Borealis of Northern Finland (https://www.ppshp.fi/Tutkimus-ja-opetus/Biopankki/Pages/Biobank-Borealis-briefly-in-English.aspx), Finnish Clinical Biobank Tampere (www.tays.fi/en-US/Research_and_development/Finnish_Clinical_Biobank_Tampere), Biobank of Eastern Finland (www.ita-suomenbiopankki.fi/en), Central Finland Biobank (www.ksshp.fi/fi-FI/Potilaalle/Biopankki), Finnish Red Cross Blood Service Biobank (www.veripalvelu.fi/verenluovutus/biopankkitoiminta), Terveystalo Biobank (www.terveystalo.com/fi/Yritystietoa/Terveystalo-Biopankki/Biopankki/) and Arctic Biobank (https://www.oulu.fi/en/university/faculties-and-units/faculty-medicine/northern-finland-birth-cohorts-and-arctic-biobank). All Finnish Biobanks are members of BBMRI.fi infrastructure (www.bbmri.fi). Finnish Biobank Cooperative-FINBB (https://finbb.fi/) is the coordinator of BBMRI-ERIC operations in Finland. The Finnish biobank data can be accessed through the Fingenious® services (https://site.fingenious.fi/en/) managed by FINBB. Equally, we want to acknowledge the participants of the EstBB for their contributions. The Estonian Genome Center analyses were partially carried out in the High-Performance Computing Center, University of Tartu. Estonian Biobank Research Team-data collection, genotyping, quality control and imputation. Andres Metspalu, Mait Metspalu, Lili Milani, Reedik Mägi, Mari Nelis, Georgi Hudjashov, Tõnu Esko. H.H. received funding for this project from Finland's Doctoral Education Pilot project. The work of the E.A. was funded by the European Union through Horizon Europe research and innovation programs under grant nos. 101137201 and 101137154, and Estonian Research Council Grant PRG1291. Open access funded by Helsinki University Library.

## Author contributions

H.M.O. and H.H. designed the study. H.H., R.E., E.A., and J.V. ran the analyses. H.H. wrote the initial manuscript. H.M.O., H.H., R.E., A.T., E.A., and J.V. contributed to reviewing and finalizing the manuscript.

## Competing interests

The authors declare no competing interests.

## Additional information

**Supplementary information** The online version contains Supplementary material available at https://doi.org/10.1038/s41467-026-69670-z.

## Estonian Biobank Research Team

Erik Abner ⓘ [4]

## FinnGen

Hele Haapaniemi ⓘ [1], Reyhane Eghtedarian[1] & Hanna M. Ollila ⓘ [1,2,3,5] ✉

