## [Transparent Peer Review File · Nature Communications]

The genetic basis of dermatophytosis skin infection susceptibility

Corresponding Author: Dr Hanna Ollila

Version 0:

Reviewer comments:

Reviewer #1

(Remarks to the Author)

Summary: The authors describe a genome-wide association study of dermatophytosis (ringworm) infection in 4 large national biobanks. Through meta-analysis, they identify 30 genome-wide significant loci including missense variants, variants in LD with missense variants and variants driving gene expression and overlapping with related conditions. Assessing the function of the genes underlying the associated loci, the authors suggest a role for altered keratin processing in dermatophytosis and, using LD score regression, the authors further show implication of connective tissue and immune cells as the primary tissues implicated in dermatophytosis. Finally, the authors demonstrate genetic overlap between dermatophytosis and several skin and immune traits as well as obesity. Overall this is a well-constructed study that improve our understanding of the genetic architecture of dermatophytosis, I have only minor comments/suggestions:

1. The analysis of missense variants as presented suggests these are causal variants at the associated loci. While this may be the case, their polyphen scores suggest they are benign and it is equally likely that there may be regulatory variants in LD with the GWAS lead SNPs that are causal. I recommend editing the text to reflect that more work needs to be done to assess the functionality of the missense variants
2. The overlap between FTO and IRX3 could be better described. I'm no an obesity researcher and am unfamiliar with the function of IRX3. Some additional description of the potential biology at this locus is warranted.
3. The section running from lines 180 to 213 comes off more as discussion points as there are very few results reported. The overall presentation of the manuscript would be improved by moving most of this to discussion and including the eQTL results in the co-localization section (presuming that a formal colocalization analysis can be done between ZNF646 and FLG).
4. Table 1 should report the effect allele. Also, given that this is a binary phenotype, I suggest reporting the effect size as Odds Ratio and 95% CI rather than beta and SE but this is personal preference.
5. Similarly in table two, when reporting association statistics the effect allele and variance around the point estimate of the effect should be reported. For this particular table, the association statistics could be moved to the supplementary materials to simplify the presentation.

Reviewer #2

(Remarks to the Author)

This study provides valuable insights into dermatophytosis and its susceptibility to skin infections, employing a genetic approach through GWAS data, which includes cohorts from the UK Biobank, the Estonian Biobank, the FinnGen project, and the Million Veterans Program. A total of 250,000 cases and 1.37 million controls were included in the meta-analysis. The researchers identified 30 genome-wide significant loci, including 9 missense variants or variants in high linkage disequilibrium (LD) with missense variants, in genes related to keratin lifecycle, skin integrity, immune defense, and obesity.

Overall, the manuscript is very interesting, the methods seem sound. However, I noticed the absence of a citation and discussion of a recent 2022 study that addresses the same topic; the researchers had also utilized the UK Biobank cohort,

but identified a different gene locus. Given that - in your study - a meta-analysis was conducted, I expect that data from each cohort, including the UK Biobank, are taken separately. The presence of such substantial differences in the gene loci identified from the same cohort raises important questions, yet these have not been discussed. Even if you might have reasons for not having included the study, the study cannot be completely neglected. Differences and potential overlaps with the 2022 study should be addressed and discussed.

Additionally, there are several points that should be clarified:

1. Introduction: Is there any specific age peak for skin infections, how about children and tinea?
2. Are there any other genetic studies available beyond the 2022 research?
3. Are there any heritability estimates known?
4. Results: Line 94 mentions an HLA locus, but it is not specified. Please clarify.
5. Given the wide variety of tinea types (e.g., skin, hair, nail infections), and considering the large cohort with comprehensive clinical data, wouldn't it be valuable to stratify the analysis based on different subtypes of tinea.
6. Discussion: Many of the identified loci are also linked to other autoimmune disorders, such as rheumatoid arthritis, alopecia areata, and atopic conditions. These are summarized under inflammation, but this might not be immediately clear to readers unfamiliar with the genetic associations of these disorders.

Version 1:

Reviewer comments:

Reviewer #1

(Remarks to the Author)

The authors have adequately responded to my suggestions. I have no further comments and support publication of the revised manuscript.

Reviewer #2

(Remarks to the Author)

I am very much satisfied with all changes the author made.

REVIEWER COMMENTS

Reviewer #1 (Remarks to the Author):

Summary: The authors describe a genome-wide association study of dermatophytosis (ringworm) infection in 4 large national biobanks. Through meta-analysis, they identify 30 genome-wide significant loci including missense variants, variants in LD with missense variants and variants driving gene expression and overlapping with related conditions. Assessing the function of the genes underlying the associated loci, the authors suggest a role for altered keratin processing in dermatophytosis and, using LD score regression, the authors further show implication of connective tissue and immune cells as the primary tissues implicated in dermatophytosis. Finally, the authors demonstrate genetic overlap between dermatophytosis and several skin and immune traits as well as obesity. Overall this is a well-constructed study that improve our understanding of the genetic architecture of dermatophytosis, I have only minor comments/suggestions:

Question 1.1 The analysis of missense variants as presented suggests these are causal variants at the associated loci. While this may be the case, their polyphen scores suggest they are benign and it is equally likely that there may be regulatory variants in LD with the GWAS lead SNPs that are causal. I recommend editing the text to reflect that more work needs to be done to assess the functionality of the missense variants.

Response 1.1 Thank you for the suggestion. We have edited the text to highlight that the lead variant can also be a regulatory variant in LD with a causal variant. We also have added that more work is needed to assess the functionality of the missense variants. This text is incorporated into the discussion as follows:

“It should be also noted that the association with a missense variant does not necessarily mean that they would be the causal variant and instead could indicate LD with a regulatory variant. Therefore, the role of the missense variants should be addressed in future functional studies.”

Question 1.2 The overlap between *FTO* and *IRX3* could be better described. I’m not an obesity researcher and am unfamiliar with the function of *IRX3*. Some additional description of the potential biology at this locus is warranted.

Response 1.2

We now describe the overlap between *FTO* and *IRX3* in the results describing its earlier reported co-localization and chromatin interaction. Additionally, we also describe *FTO* locus in obesity in the discussion.

Results: “Although obesity-associated variants are located within the *FTO* gene, they regulate the expression of the distant *IRX3* gene through long-range chromatin interactions explaining the colocalization signal at this gene region¹⁵. “

Discussion: “*FTO* locus is a well-known genetic region responsible for energy homeostasis and body weight. Variation in its intronic region are among the strongest associations with obesity¹⁵. The functional effects of *FTO* variants are mediated through homeobox genes including *IRX3* and the variants at the *FTO* locus act as long-range enhancers¹⁵.”

Question 1.3 The section running from lines 180 to 213 comes off more as discussion points as there are very few results reported. The overall presentation of the manuscript would be improved by moving most of this to discussion and including the eQTL results in the co-localization section (presuming that a formal colocalization analysis can be done between *ZNF646* and *FLG*).

Response 1.3 Thank you for your comment. This is the chapter “GWAS associations highlight the role of compromised keratin processing behind dermatophytosis”. In order to provide the reader the context of results following this paragraph, we feel that grouping the genes into three different categories would be beneficial and facilitate understanding of the following sections.

We did not perform a formal colocalization analysis for the missense variants, as they are likely located within the causal gene itself.

Question 1.4 Table 1 should report the effect allele. Also, given that this is a binary phenotype, I suggest reporting the effect size as Odds Ratio and 95% CI rather than beta and SE but this is personal preference.

Response 1.4.

We have added the OR (95% CI) as a column in Table 1 and 2. Additionally, we have also added effect and reference alleles to both tables.

Question 1.5 Similarly in table two, when reporting association statistics the effect allele and variance around the point estimate of the effect should be reported. For this particular table, the association statistics could be moved to the supplementary materials to simplify the presentation.

Response 1.5

We have added effect and reference alleles as well as confidence intervals around the point estimate (OR 95 % CI) in Table 2 and moved beta and se to supplementary material.

Reviewer #2 (Remarks to the Author):

This study provides valuable insights into dermatophytosis and its susceptibility to skin infections, employing a genetic approach through GWAS data, which includes cohorts from the UK Biobank, the Estonian Biobank, the FinnGen project, and the Million Veterans Program. A total of 250,000 cases and 1.37 million controls were included in the meta-analysis. The researchers identified 30 genome-wide significant loci, including 9 missense variants or variants in high linkage disequilibrium (LD) with missense variants, in genes related to keratin lifecycle, skin integrity, immune defense, and obesity.

Overall, the manuscript is very interesting, the methods seem sound. However, I noticed the absence of a citation and discussion of a recent 2022 study that addresses the same topic; the researchers had also utilized the UK Biobank cohort, but identified a different gene locus. Given that - in your study - a meta-analysis was conducted, I expect that data from each cohort, including the UK Biobank, are taken separately. The presence of such substantial differences in the gene loci identified from the same cohort raises important questions, yet these have not been discussed. Even if you might have reasons for not having included the study, the study cannot be completely neglected. Differences and potential overlaps with the 2022 study should be addressed and discussed.

Response to summary: Thank you for the overall comments and particularly pointing out the UKB paper. The earlier paper from UKB used Hospital diagnosis data only which contains 630 individuals with Dermatophytosis. We used additionally the data from primary care that includes 30,000 individuals with dermatophytosis diagnosis [Figure R1]. The difference in the cohorts for number of individuals and number of associating loci stem from the large difference in the number of cases that were not included in the earlier study.

Additionally, the earlier described variant does not have correlated variants in the regional association, and does not show any association with Dermatophytosis in our cohort ($P = 0.67$). This makes it possible that the earlier described association is false positive driven by the small number of individuals in the earlier GWAS and the low allele frequency of the risk TINAG variant.

To acknowledge the earlier effort, we have included the result from the paper in the discussion including a citation to the paper.

Figure R1. *Number of individuals with dermatophytosis from UKB registry data. The difference between the 2022 publication and our work is likely the number of cases. Earlier work used hospital level data only (N = 616), whereas our study also uses primary care data (N=32,414). It is possible that additional differences are related to the clinical picture, but this is unlikely given the relatively small sample size by the earlier publication.*

Additionally, there are several points that should be clarified:

Question 2.1. Introduction: Is there any specific age peak for skin infections, how about children and tinea?

Response 2.1 We added the following information in the introduction: “Dermatophytosis skin infections can develop to people of all ages. Infections of the scalp (tinea capitis), are more common in children, while other tineaes more commonly affect postpubertal individuals with a peak at mid-age.^{7,8}”

Question 2.2. Are there any other genetic studies available beyond the 2022 research?

Response 2.2. We did not find other genome-wide association studies on dermatophytosis.

Question 2.3. Are there any heritability estimates known?

Response 2.3. Earlier twin studies or GWAS have not estimated the heritability of dermatophytosis. To fill this gap in the knowledge we computed the snp-based heritability from the current meta-analysis summary statistics as well as for all dermatophytosis subtypes (ICD-10 B35.x) in FinnGen. The heritabilities were generally low (h^2 meta-analysis = 0.0075). These heritabilities align with overall infection disease heritabilities where the environmental exposure to infection to start with typically has a large impact. For example, COVID-19 heritability was at a similar scale (h^2 = 0.01). However, it does not necessarily mean that there would be only a small number of associating variants. We provide the heritability estimates in Table S8.

Question 2.4. Results: Line 94 mentions an HLA locus, but it is not specified. Please clarify.

Response 2.4 We have additionally performed a correlation analysis between our lead variant rs1794269 and all HLA alleles with prevalence over 1 % in FinnGen study. We show a strongest correlation with DQB1*05:01 with $r = 0.48$. In addition we performed a formal HLA fine mapping in FinnGen for all HLA haplotypes. This analysis supports the association of DQB1*05:01 with Dermatophytosis with p-value of 0.005 and $\beta = 0.03$.

We have edited the text:

“The lead variant for the association (rs1794269, $\beta = 0.046$ and $p = 1.42 \times 10^{-36}$) was located closest to the *HLA-DQB1* gene but a correlation analysis suggests a strongest correlation with DQB1*05:01 with $r = 0.48$. Formal HLA fine mapping in FinnGen supports the association of DQB1*05:01 with Dermatophytosis with p-value of 0.005. The association between *HLA* region and dermatophytosis highlights an overall immune signal in dermatophytosis.”

Question 2.5. Given the wide variety of tinea types (e.g., skin, hair, nail infections), and considering the large cohort with comprehensive clinical data, wouldn't it be valuable to stratify the analysis based on different subtypes of tinea.

Response 2.5 We have run GWAS analyses separately for each subtype of dermatophytosis (B35.1, B35.2, B35.3, B35.4, B35.4, B35.5, B35.6, B35.8, and B35.9.) using data from FinnGen, Estonian Biobank, MVP and UK Biobank. We report the association statistics of main dermatophytosis lead variants in each subtype meta-analysis (Supplementary excel S9) and show that many of these are seen also in specific subtypes especially in B35.1 Tinea unguium (nail ringworm) and B35.3 Tinea pedis (athlete's foot), which have most cases and controls.

Additionally, we report novel subtype specific associations in the manuscript.

We have added the following results to the main manuscript: "Dermatophytosis can be classified into subtypes based on the region of the body where the infection is manifesting. To understand the subtype specific associations we additionally ran GWAS on each of these subtypes in FinnGen, EstBB, UKB, along with the publicly available genome-wide association data for subtypes in the MVP cohort (Figure 2). The association statistics for each of our lead variants in these subtypes are reported in Supplementary Table S9 (case and control numbers are presented in S8). We show association of 10 lead variants from the main dermatophytosis analysis at the subtype level ($p < 0.0002$, Bonferroni corrected threshold) especially in subtypes B35.1 (nail ringworm) and B35.3 (athlete's foot). When studying the lead variants at the subtype level, we see a consistent pleiotropic association with a subset of variants across several different subtypes; *FLG* (B35.1 and B35.3), *HLA* (B35.1), *LYNX1* (B35.3) and *FTO* (B35.1) at genome-wide significant level ($p < 5 \times 10^{-8}$) highlighting the role of these genes in the defence against dermatophytosis infections. In addition, we report novel associations in another zinc finger gene *ZNF668* (B35.1 and B35.3) located in proximity of the *ZNF646* gene. Furthermore we report 24 novel associations, such as rs12091247 closest to *NOTCH2* gene that is linked to skin homeostasis and epidermal differentiation. We present Manhattan plots for each subtype in Supplementary Figures and the lead variants in Supplementary Table S10."

Question 2.6. Discussion: Many of the identified loci are also linked to other autoimmune disorders, such as rheumatoid arthritis, alopecia areata, and atopic conditions. These are summarized under inflammation, but this might not be immediately clear to readers unfamiliar with the genetic associations of these disorders.

Response 2.6. This is a great point, and we have clarified the connection between autoimmune diseases and dermatophytosis: "While dermatophytosis is primarily an infection, the variants that were associated with it, were also associated with autoimmune traits. The overall connection between infectious and immune loci that associate with autoimmunity needs to be further explored in future studies."